# Chromosome Painting in Cultivated Bananas and Their Wild Relatives (*Musa* spp.) Reveals Differences in Chromosome Structure

**DOI:** 10.3390/ijms21217915

**Published:** 2020-10-24

**Authors:** Denisa Šimoníková, Alžběta Němečková, Jana Čížková, Allan Brown, Rony Swennen, Jaroslav Doležel, Eva Hřibová

**Affiliations:** 1Institute of Experimental Botany of the Czech Academy of Sciences, Centre of the Region Hana for Biotechnological and Agricultural Research, 77900 Olomouc, Czech Republic; simonikova@ueb.cas.cz (D.Š.); nemeckova@ueb.cas.cz (A.N.); cizkova@ueb.cas.cz (J.Č.); dolezel@ueb.cas.cz (J.D.); 2International Institute of Tropical Agriculture, Banana Breeding, PO Box 447 Arusha, Tanzania; A.Brown@cgiar.org (A.B.); R.Swennen@cgiar.org (R.S.); 3Division of Crop Biotechnics, Laboratory of Tropical Crop Improvement, Katholieke Universiteit Leuven, 3001 Leuven, Belgium

**Keywords:** chromosome translocation, fluorescence in situ hybridization, karyotype evolution, oligo painting FISH, structural chromosome heterozygosity

## Abstract

Edible banana cultivars are diploid, triploid, or tetraploid hybrids, which originated by natural cross hybridization between subspecies of diploid *Musa acuminata*, or between *M. acuminata* and diploid *Musa balbisiana*. The participation of two other wild diploid species *Musa schizocarpa* and *Musa textilis* was also indicated by molecular studies. The fusion of gametes with structurally different chromosome sets may give rise to progenies with structural chromosome heterozygosity and reduced fertility due to aberrant chromosome pairing and unbalanced chromosome segregation. Only a few translocations have been classified on the genomic level so far, and a comprehensive molecular cytogenetic characterization of cultivars and species of the family Musaceae is still lacking. Fluorescence in situ hybridization (FISH) with chromosome-arm-specific oligo painting probes was used for comparative karyotype analysis in a set of wild *Musa* species and edible banana clones. The results revealed large differences in chromosome structure, discriminating individual accessions. These results permitted the identification of putative progenitors of cultivated clones and clarified the genomic constitution and evolution of aneuploid banana clones, which seem to be common among the polyploid banana accessions. New insights into the chromosome organization and structural chromosome changes will be a valuable asset in breeding programs, particularly in the selection of appropriate parents for cross hybridization.

## 1. Introduction

Banana represents one of the major staple foods and is one of the most important cash crops with the estimated value of $25 billion for the banana industry. The annual global production of bananas reached 114 million tons in 2017 [1], with about 26 million tons exported in 2019 [2]. Two types of bananas are known—sweet bananas, serving as a food supplement, and cooking bananas, which are characterized by starchier fruits [3]. Edible banana cultivars are vegetatively propagated diploid, triploid, and tetraploid hybrids, which originated after natural cross hybridization between the wild diploids *Musa acuminata* (2n = 2x = 22, AA genome) and *Musa balbisiana* (2n = 2x = 22, BB genome) and their hybrid progenies. To some extent, other *Musa* species such as *Musa schizocarpa* (2n = 2x = 22, SS genome) and *Musa textilis* (2n = 2x = 20, TT genome) also contributed to the origin of some edible banana clones [4,5,6].

Based on morphology and geographical distribution, *M. acuminata* has been divided into nine subspecies (*banksii, burmannica, burmannicoides, errans, malaccensis, microcarpa, siamea, truncata,* and *zebrina*) and three varieties (chinensis, sumatrana, and tomentosa) [7,8,9]. It has been estimated that at least four subspecies of *M. acuminata* contributed to the origin of cultivated bananas [7,10]. Out of them, *M. acuminata* ssp. *banksii*, with the original center of diversity in New Guinea played a major role in this process [6,11,12]. Other subspecies were *M. acuminata* ssp. *burmannica* with the center of diversity in Myanmar [13]; ssp. *malaccensis* with the origin in Malay peninsula [7,14]; and ssp. *zebrina,* which originated in Indonesia [10].

It is believed that human migration together with a different geography of the present archipelago in Southeast Asia during the glacial period, when a drop in the sea level resulted in the interconnection of current islands in Southeast Asia into one land mass [15,16,17,18], brought different *M. acuminata* subspecies into close proximity, enabling cross hybridization and giving rise to diploid intraspecific hybrids that were subjected to human selection and propagation [7,8]. The fusion of unreduced gametes produced by diploid edible and partially sterile cultivars with normal haploid gametes from fertile diploids [4,19,20] would give rise to triploids. The number one export dessert banana type Cavendish as well as other important dessert bananas, such as Gros Michel and Pome types, originated according to this scenario by the hybridization of a diploid representative of subgroup “Mchare” (originally named “Mlali”) (AA genome; *zebrina/microcarpa* and *banksii* ascendance), which served as a donor of an unreduced diploid gamete with the haploid gamete of “Khai” (*malaccensis* ascendance) [12,21,22].

Another group of edible triploid bananas, clones with AAB (so called plantains) or ABB constitution, cover nearly 40% of global banana production, whereas plantains stand for 18% of total banana production [23]. These interspecific triploid cultivars originated after fusion of an unreduced gamete from interspecific AB hybrid with haploid gamete from diploid *M. acuminata* ssp. *banksii* or *M. balbisiana* [7,23]. Their evolution most probably involved several backcrosses [24]. Two important AAB subgroups of starchy bananas evolved in two centers of diversity, the African Plantains and the Pacific (Maia Maoli/Popoulu) Plantains [14].

East African Highland bananas (EAHB) represent an important starchy type of banana for over 80 million people living in the Great Lakes region of East Africa, which is considered a secondary center of banana diversity [25,26]. EAHBs are a subgroup of triploid bananas with AAA constitution, which arose from hybridization between diploid *M. acuminata* ssp. *banksii* and *M. acuminata* ssp. *zebrina*. However, *M. schizocarpa* also contributed to the formation of these hybrids [6]. Interestingly, different EAHB varieties have relatively low genetic diversity, contrary to morphological variation, which occurred most probably due to the accumulation of somatic mutations and epigenetic changes [6,7,12,27,28].

Cultivated clones originating from inter-subspecific and interspecific hybridization and with the contribution of unreduced gametes in the case of triploid clones have reduced or zero production of fertile gametes. This is a consequence of aberrant chromosome pairing during meiosis due to structural chromosome heterozygosity and/or odd ploidy levels. Reduced fertility greatly hampers the efforts to breed improved cultivars [8,23,29,30], which are needed to satisfy the increasing demand for dessert and starchy bananas under the conditions of climate change and increasing pressure of pests and diseases [28].

Traditional breeding strategies for triploid banana cultivars involve the development of tetraploids (4x) from 3x × 2x crosses, followed by the production of secondary triploid hybrids (3x) from 4x × 2x crosses [31,32,33,34]. Similarly, diploids also play important role in the breeding strategy of diploid cultivars, 4x × 2x crosses are used to create improved cultivars [33]. It is thus necessary to identify cultivars that produce seeds under specific conditions, followed by breeding for target traits and re-establishing seed-sterile end products. Low seed yield after pollination (e.g., 4 seeds per Matooke (genome AAA) bunch and only 1.6 seeds per Mchare (genome AA) bunch followed by embryo rescue, with very low germination rate (~2%) illustrates the serious bottleneck for the breeding processes (Brown and Swennen, unpublished).

Since banana breeding programs use diploids as the principal vehicle for introducing genetic variability (e.g., [35,36,37]), the knowledge of their genome structure at chromosomal level is critical to reveal possible causes of reduced fertility and the presence of non-recombining haplotype blocks, to provide data to identify the parents of cultivated clones that originated spontaneously without direct human intervention, and to select parents for improvement programs. In order to provide insights into the genome structure at chromosome level, we employed fluorescence in situ hybridization (FISH) with chromosome-arm-specific oligo painting probes in a set of wild *Musa* species and edible banana clones potentially useful in banana improvement. Chromosome painting in twenty representatives of the Eumusa section of genus *Musa*, which included subspecies of *M. acuminata*, *M. balbisiana,* and their inter-subspecific and interspecific hybrids, revealed chromosomal rearrangements discriminating subspecies of *M. acuminata* and structural chromosome heterozygosity of cultivated clones. Identification of chromosome translocations pointed to particular *Musa* subspecies as putative parents of cultivated clones and provided an independent support for hypotheses on their origin.

## 2. Results

### 2.1. Karyotype of *M. acuminata* and *M. balbisiana*

The first part of the study focused on comparative karyotype analysis in six subspecies of *M. acuminata* and in *M. balbisiana*. Oligo painting FISH in structurally homozygous *M. acuminata* ssp. *banksii* “Banksii” and *M. acuminata* ssp. *microcarpa* “Borneo” did not reveal detectable chromosome translocations (Figure 1A) as compared to the reference banana genome of *M. acuminata* ssp. *malaccensis* “DH Pahang” [38]. Thus, the three subspecies share the same overall organization of their chromosome sets. On the other hand, different chromosome translocations, which were found to be subspecies specific, were identified in the remaining four subspecies of *M. acuminata* (ssp. *zebrina*, ssp. *burmannicoides*, ssp. *siamea,* ssp. *burmannica*).

In *M. acuminata* ssp. *zebrina*, a reciprocal translocation between the short arm of chromosome 3 and the long arm of chromosome 8 was identified (Figure 1B, Appendix A). Three phylogenetically closely related subspecies *M. acuminata* ssp. *burmannicoides*, *M. acuminata* ssp. *siamea,* and *M. acuminata* ssp. *burmannica* shared two translocations (Figure 1C–E, Appendix A). The first of these involved the transfer of a segment of the long arm of chromosome 8 to the long arm of chromosome 2; the second was a reciprocal translocation involving a large segment of the short arm of chromosome 9 and the long arm of chromosome 1. In addition to the translocations shared by representatives of the three subspecies, chromosome painting revealed additional subspecies-specific translocations.

In *M. acuminata* ssp. *siamea* “Pa Rayong”, translocation of a small segment of the long arm of chromosome 3 to the short arm of chromosome 4 was detected. Importantly, this translocation was visible only on one of the homologs, indicating structural chromosome heterozygosity and a hybrid origin (Figure 1D, Appendix A). Subspecies-specific translocations involving only one of the homologs were also found in *M. acuminata* ssp. *burmannica* “Tavoy”. They included a reciprocal translocation between chromosomes 3 and 8, which was also detected in *M. acuminata* ssp. *zebrina,* and a Robertsonian translocation between chromosomes 7 and 8, which gave rise to a chromosome comprising the long arms of chromosomes 7 and 8 and a chromosome made of the short arms of chromosomes 7 and 8 (Figure 1E, Appendix A).

In *M. balbisiana* “Pisang Klutuk Wulung” translocation of a small segment of the long arm of chromosome 3 to the long arm of chromosome 1 was observed (Figure 1F), similar to the translocation identified in our previous work [38] in *M. balbisiana* “Tani”. In agreement with a low level of genetic diversity of *M. balbisiana* [18,39], no detectable differences in karyotypes were found between both accessions of the species.

### 2.2. Karyotype Structure of Edible Clones That Originated as Intraspecific Hybrids

Chromosome painting in diploid cooking banana cultivars belonging to the Mchare group with AA genome, confirmed their hybrid origin. All three accessions analyzed in this work comprised two reciprocal translocations, with each of them observed only in one chromosome set. The first translocation involved the short segments of the long arms of chromosomes 4 and 1 (Figure 2A), while the second one involved a reciprocal translocation between the short arm of chromosome 3 and the long arm of chromosome 8 (Figure 2A, Appendix A). Both translocations were observed in a heterozygous state in all three analyzed Mchare banana representatives. Reciprocal translocation involving short segments of the long arms of chromosomes 4 and 1 was also identified in diploid cultivar “Pisang Lilin” with AA genome (Figure 2B, Appendix A). This clone is used in breeding programs as a donor of useful agronomical characteristics, such as female fertility or resistance to yellow Sigatoka [40]. Moreover, this accession was heterozygous for the translocation, indicating a hybrid origin (Figure 2B).

Two representatives of triploid dessert banana cultivars “Cavendish” and “Gros Michel” with AAA genome constitution displayed identical chromosome structure as assessed by chromosome painting. One of the three chromosome sets was characterized by reciprocal translocation between short segments of the long arms of chromosomes 4 and 1 (3A, Appendix A). It is worth mentioning that the same translocation was identified in Mchare cultivars and in “Pisang Lilin” (Figure 2A,B). Another chromosome set of “Cavendish” and “Gros Michel” (Figure 3A, Appendix A) contained a reciprocal translocation between the short arm of chromosome 3 and the long arm of chromosome 8, which was also identified in *M. acuminata* ssp. *zebrina* (Figure 1B). In addition to the two translocations, a translocation of the short arm of chromosome 7 to the long arm of chromosome 1 was observed in both accessions of desert banana. The translocation resulted in the formation of a small telocentric chromosome consisting of only the long arm of chromosome 7 (Figure 3A, Appendix A).

East African Highland bananas (EAHB) represent an important group of triploid cultivars with the genome constitution AAA. We have analyzed two accessions of these cooking bananas. Both cultivars (“Imbogo” and “Kagera”) contained a reciprocal translocation between the short arm of chromosome 3 and the long arm of chromosome 8, which was also observed in *M. acuminata* ssp. *zebrina.* This translocation was identified only in two out of the three chromosome sets (Figure 3B,C, Appendix A). Karyotype analysis revealed that cultivar “Imbogo” lacked one chromosome and was aneuploid (2n = 32). Chromosome painting facilitated identification of the missing chromosome and suggested the origin of the aneuploid karyotype (Figure 3C, Appendix A), which involved a Robertsonian translocation between chromosomes 7 and 1, giving rise to a recombined chromosome containing the long arms of chromosomes 7 and 1. Our observation suggests that the short arms of the two chromosomes were lost (Figure 3C, Appendix A). The loss of a putative chromosome comprising two short arms is a common consequence of the Robertsonian translocation [41].

### 2.3. Karyotype Structure of Interspecific Banana Clones

Plantains are an important group of starchy type of bananas with AAB genome constitution and originated as hybrids between *M. acuminata* (A genome) and *M. balbisiana* (B genome). Chromosome painting in three cultivars representing two plantain morphotypes-Horn (cultivar “3 Hands Planty”) and French type (cultivars “Obino l’Ewai” and “Amou”) confirmed the presence of B-genome-specific translocation in one chromosome set in all three accessions, i.e., the translocation of a small segment of the long arm of chromosome 3 to the long arm of chromosome 1 (Figure 3D,E, Appendix A). No other translocation was found in these cultivars. However, the clone “Amou” was found to be aneuploid (2n = 32) as it missed one copy of chromosome 2 (Figure 3E, Appendix A).

In agreement with their predicted ABB genome constitution, B-genome-specific chromosome translocation was also observed in two chromosome sets in triploid cultivars “Pelipita” and “Saba sa Hapon” (Figure 3F, Appendix A). No other translocation was found in these cultivars.

## 3. Discussion

Until recently, cytogenetic analysis in plants was hindered by the lack of available DNA probes suitable for fluorescent painting of individual chromosomes [42,43]. The only option was to use pools of bacterial artificial chromosome (BAC) clones, which were found useful in plant species with small genomes (e.g., [44,45]). The development of oligo painting FISH [46] changed this situation dramatically, and it is now possible to label individual plant chromosomes and chromosomal regions in many species [47,48,49,50,51,52]. Chromosome-arm-specific oligo painting probes were also recently developed for banana (*Musa* spp.) by Šimoníková et al. [38], who demonstrated the utility of this approach for anchoring DNA pseudomolecules of a reference genome sequence to individual chromosomes in situ and for the identification of chromosome translocations.

In this work, we used oligo painting FISH for comparative karyotype analysis in a set of *Musa* accessions comprising wild species used in banana breeding programs and economically significant edible cultivars (Appendix A). These experiments revealed chromosomal translocations in subspecies of *M. acuminata* (A genome), their intraspecific hybrids as well as in *M. balbisiana* (B genome) and in interspecific hybrid clones originating from cross hybridization between *M. acuminata* and *M. balbisiana* (Figure 4). A difference in chromosome structure among *M. acuminata* subspecies was suggested earlier by Shepherd [53], who identified seven translocation groups in *M. acuminata* based on chromosome pairing during meiosis. An independent confirmation of this classification was the observation of segregation distortion during genetic mapping in inter-subspecific hybrids of *M. acuminata* [54,55,56,57].

### 3.1. Structural Genome Variation in Diploid *M. acuminata*

In this work, we analyzed one representative of each of six subspecies of *M. acuminata*. We cannot exclude differences in chromosome structure within individual subspecies. However, given the large genetic homogeneity of the six subspecies as clearly demonstrated by molecular markers [6,12,28,58,59,60], this does not seem probable. We observed a conserved genome structure in *M. acuminata* ssp. *banksii* and *M. acuminata* ssp. *microcarpa,* which did not contain any translocation chromosome when compared to the reference genome of *M. acuminata* ssp. *malaccensis* “DH Pahang” [61]. The genome structure shared by the three subspecies was also observed in *M. schizocarpa* [38] and corresponds to the standard translocation (ST) group as defined by Shepherd [53].

The other chromosome translocation group defined by Shepherd [53], the Northern Malayan group (NM), is characteristic for *M. acuminata* ssp. *malaccensis* and some AA cultivars, including “Pisang Lilin”. Our results revealed a reciprocal translocation between chromosomes 1 and 4 in one chromosome set of “Pisang Lilin”, thus confirming Shepherd’s characterization of this clone as heterozygous, having ST x NM genome structure. Before the *Musa* genome sequence became available, Hippolyte et al. [55] assumed the presence of a duplication of the distal region of chromosome 1 on chromosome 4 in this clone based on comparison of high-density genetic maps. Taking the advantage of the availability of a reference genome sequence and after resequencing genomes of a set of *Musa* species, Martin et al. [8] described heterozygous reciprocal translocation between chromosomes 1 and 4, involving 3 Mb of the long arm of chromosome 1 and a 10 Mb segment of the long arm of chromosome 4 in *M. acuminata* ssp. *malaccensis*. Further experiments indicated preferential transmission of the translocation to the progeny and its frequent presence in triploid banana cultivars [8]. However, it is not clear whether the reciprocal translocation between chromosomes 1 and 4, which we observed in the heterozygous state only, originated in ssp. *malaccensis*, or whether it was transmitted to genomes of some *malaccensis* accessions by ancient hybridization events [8]. Three phylogenetically closely related subspecies *M. acuminata* ssp. *burmannica*, *M. acuminata* ssp. *burmannicoides,* and *M. acuminata* ssp. *siamea*, which have similar phenotype and geographic distribution [12,19] share a translocation of a part of the long arm of chromosome 8 to the long arm of chromosome 2, and a reciprocal translocation between chromosomes 1 and 9. These translocations were identified recently by Dupouy et al. [60] after mapping mate-paired Illumina sequence reads to the reference genome of “DH Pahang”. The authors estimated the size of translocated region of chromosome 8 to chromosome 2 to be 7.2 Mb, while the size of the distal region of chromosome 2, which was found translocated to chromosome 8 in wild diploid clone “Calcutta 4” (ssp. *burmannicoides*) was estimated to be 240 kb [60]. The size of the translocated regions of chromosomes 1 and 9 was estimated to be 20.8 and 11.6 Mb, respectively [60]. Using oligo painting, we did not detect the 240 kb distal region of chromosome 2 translocated to chromosome 8, and this may reflect the limitation in the sensitivity of whole chromosome arm oligo painting.

The shared translocations in all three subspecies of *M. acuminata* (*burmannicoides*, *burmannica,* and *siamea*) support their close phylogenetic relationship as proposed by Shepherd [53] and later verified by molecular studies [6,12,28,58,59]. Dupouy et al. [60] coined the idea of a genetically uniform *burmannica* group. However, our data indicate a more complicated evolution of the three genotypes recognized as representatives of different *acuminata* subspecies. First, the characteristic translocations between chromosomes 2 and 8, and 1 and 9 were detected only on one chromosome set in *burmannica* “Tavoy”, as compared to those in *M. acuminata* “Calcutta 4” (ssp. *burmannicoides*) and “Pa Rayong” (ssp. *siamea*). Second, we observed two subspecies-specific translocations in ssp. *burmannica* only in one chromosome set, and we detected additional subspecies-specific translocation in *M. acuminata* ssp. *siamea* only in one chromosome set, indicating its hybrid origin. Based on these results, we hypothesize that ssp. *burmannicoides* could be a progenitor of the clones characterized by structural chromosome heterozygosity. Divergence in genome structure between the three subspecies (*burmannica, burmannicoides,* and *siamea*) was demonstrated also by Shepherd [53], who classified some *burmannica* and *siamea* accessions as Northern 2 translocation group of *Musa*, differing from the Northern 1 group (*burmannicoides* and other *siamea* accessions) by one additional translocation.

We observed subspecies-specific translocations also in *M. acuminata* ssp. *zebrina* “Maia Oa”. In this case, chromosome painting revealed a Robertsonian translocation between chromosomes 3 and 8. Interestingly, Dupouy et al. [60] failed to detect this translocation after sequencing mate-pair libraries using Illumina technology. The discrepancy may point to the limitation of the sequencing approach to identify translocations arising by a breakage of (peri)centromeric regions. As these regions comprise mainly DNA repeats, they may not be assembled properly in a reference genome, thus preventing their identification by sequencing. In fact, this problem may also be encountered if subspecies-specific genome regions are absent in the reference genome sequence. We also revealed the *zebrina*-type translocation (a reciprocal translocation between chromosomes 3 and 8) in all three analyzed cultivars of diploid Mchare banana. The presence of a translocation between the long arm of chromosome 1 and the long arm of chromosome 4 on one chromosome set of Mchare indicates a hybrid origin of Mchare, with ssp. *zebrina* being one of the progenitors of this banana group. This agrees with the results obtained by genotyping using molecular markers [28]. Most recently, the complex hybridization scheme of Mchare bananas was also supported by the application of transcriptomic data for the identification of specific single-nucleotide polymorphisms (SNPs) in 23 *Musa* species and edible cultivars [22].

### 3.2. Genome Structure and Origin of Cultivated Triploid *Musa* Clones

Plantains are an important group of triploid starchy bananas with AAB genome constitution, which originated after hybridization between *M. acuminata* and *M. balbisiana*. As expected, chromosome painting in “3 Hands Planty”, “Amou”, and “Obino l’Ewai” cultivars revealed B-genome-specific translocation of 8 Mb segment from the long arm of chromosome 3 to the long arm of chromosome 1. Unlike the B-genome chromosome set, the two A-genome chromosome sets of plantains lacked any detectable translocation. Genotyping using molecular markers revealed that the A genomes of the plantain group are related to *M. acuminata* ssp. *banksii* [18,58,62,63]. This is in line with the absence of chromosome translocations we observed in *M. acuminata* ssp. *banksii.*

Triploid cultivar “Pisang Awak”, a representative of the ABB group, is believed to contain one A-genome chromosome set closely related to *M. acuminata* ssp. *malaccensis* [12]. However, after simple sequence repeat (SSR) genotyping, Christelová et al. [28] found, that some ABB clones from Saba and Bluggoe-Monthan groups clustered together with the representatives of Pacific banana Maia Maoli/Popoulu (AAB). These results point to *M. acuminata* ssp. *banksii* as their most probable progenitor. Unfortunately, in this case, chromosome painting did not bring useful hints on the nature of the A subgenomes in these interspecific hybrids, as *M. acuminata* ssp. *banksii* and ssp. *malaccensis* representatives do not differ in the presence of specific chromosome translocation.

The presence of B-genome-specific translocation of a small region of long arm of chromosome 3 to the long arm of chromosome 1 [38], observed in all *M. balbisiana* accessions and their interspecific hybrids with *M. acuminata*, seems to be a useful cytogenetic landmark of the presence of B subgenome. In our study, the number of chromosome sets containing B-genome-specific translocation agreed with the predicted genomic constitution (AAB or ABB) of the hybrids. Clearly, one B-genome-specific landmark is not sufficient for the analysis of the complete genome structure of interspecific hybrids at the cytogenetic level. Further work is needed to identify additional cytogenetic landmarks to uncover the complexities of genome evolution after interspecific hybridization in *Musa*. Recently, Baurens et al. [23] analyzed the genome composition of banana interspecific hybrid clones using whole-genome sequencing strategies followed by bioinformatic analysis based on A- and B-genome-specific SNP calling and they also detected the B-genome-specific translocation in interspecific hybrids.

Chromosome painting confirmed a small genetic difference between triploid clones “Gros Michel” and “Cavendish” (AAA genomes) as previously determined by various molecular studies [20,28]. Both clones share the same reciprocal translocation between the long arms of chromosomes 1 and 4 in one chromosome set. Interestingly, diploid Mchare cultivars also contain the same translocation in one chromosome set. The presence of the translocation was also identified by sequencing genomic DNA both in “Cavendish” and “Gros Michel”, as well as in Mchare banana “Akondro Mainty” [8]. These observations confirm a close genetic relationship between both groups of edible bananas as noted previously [12,20]. According to Martin et al. [8,22], a 2n gamete donor, which contributed to the origin of dessert banana clones with AAA genomes, including “Cavendish” and “Gros Michel”, belongs to the Mchare (Mlali) subgroup. The genome of this ancient subgroup, which probably originated somewhere around Java, Borneo, and New Guinea, but today is only found in East Africa, is based on *zebrina/microcarpa* and *banksii* subspecies [12].

The third chromosome set in triploid “Cavendish” and “Gros Michel” contains a reciprocal translocation between chromosomes 3 and 8, which was detected by oligo painting FISH in the diploid *M. acuminata* ssp. *zebrina*. Our observations indicate that heterozygous representatives of *M. acuminata* ssp. *malaccensis* and *M. acuminata* spp. *zebrina* contributed to the origin of “Cavendish” and “Gros Michel” as their ancestors as suggested earlier [12,28,64]. According to Perrier et al. [12], *M. acuminata* ssp. *banksii* was one of the two progenitors of “Cavendish”/“Gros Michel” group of cultivars. However, using chromosome-arm-specific oligo painting, we observed a translocation of the short arm of chromosome 7 to the long arm of chromosome 1, resulting in a small telocentric chromosome made only of the long arm of chromosome 7, which was, up to now, identified only in these cultivars. The translocation, which gave rise to the small telocentric chromosome, could be a result of processes accompanying the evolution of this group of triploid AAA cultivars. Alternatively, another wild diploid clone, possibly structurally heterozygous, was involved in the origin of “Cavendish”/“Gros Michel” bananas.

We observed reciprocal translocation between chromosomes 3 and 8, which is typical for *M. acuminata* ssp. *zebrina*, in the economically important group of triploid East African Highland bananas (EAHB) with AAA genome constitution. An important role of *M. acuminata* ssp. *zebrina* and *M. acuminata* ssp. *banksii* as the most probable progenitors of EAHB was suggested previously [6,22,27,28,58,65]. Our results, which indicate that EAHB contained two chromosome sets from ssp. *zebrina* and one chromosome set from ssp. *banksii*, point to the most probable origin of EAHB. Hybridization between *M. acuminata* ssp. *zebrina* and ssp. *banksii* could give rise to an intraspecific diploid hybrid with reduced fertility. Triploid EAHB cultivars then could originate by backcross of the intraspecific hybrid (a donor of non-reduced gamete) with *M. acuminata* ssp. *zebrina*, or with another diploid, most probably a hybrid of *M. acuminata* ssp. *zebrina*. Moreover, phylogenetic analysis of Němečková et al. [6] surprisingly also revealed a possible contribution of *M. schizocarpa* to EAHB formation, thus indicating a more complicated origin. Further investigation is needed, and the availability of EAHB genome sequence in particular, to shed more light on the origin and evolution of these triploid clones.

### 3.3. The Origin of Aneuploidy

To date, aneuploidy in *Musa* has been identified by chromosome counting [5,6,66,67,68,69]. Although this approach is laborious and low throughput, it cannot be replaced by flow cytometric estimation of nuclear DNA amounts because of the differences in genome size between *Musa* species, subspecies, and their hybrids (e.g., [5,6,28]). A more laborious approach to achieve high-resolution flow cytometry as used by Roux et al. [70] is too slow and laborious to be practical. Thus, traditional chromosome counting [5,6,66,67,68,69] remains the most reliable approach. Obviously, it is not suitable to identify the chromosome(s) involved in aneuploidy and the origin of the aberrations.

Here, we employed chromosome painting to shed light on the nature of aneuploids among the triploid *Musa* accessions. One of the aneuploid clones was identified in the plantain “Amou”, in which one copy of chromosome 2 was lost. The origin of aneuploidy in the clone “Imbogo” (AAA genome), a representative of EAHB, involved structural chromosome changes involving the breakage of chromosomes 1 and 7 in centromeric regions, followed by the fusion of the long arms of chromosomes 1 and 7 and the subsequent loss of the short arms of both chromosomes (Appendix A). It needs to be noted that these plants were obtained from the International *Musa* Germplasm Transit Center (ITC, Leuven, Belgium), where the clones are stored in vitro. The loss of the whole chromosome in plantain “Amou” could occur during long-term culture.

To conclude, the application of oligo painting FISH improved the knowledge on genomes of cultivated banana and their wild relatives at chromosomal level. For the first time, a comparative molecular cytogenetic analysis of twenty representatives of the Eumusa section of genus *Musa*, including accessions commonly used in banana breeding, was performed using chromosome painting. However, as only a single accession from each of the six subspecies of *M. acuminata* was used in this study, our results will need to be confirmed by analyzing more representatives from each taxon. The identification of chromosome translocations pointed to particular *Musa* subspecies as putative parents of cultivated clones and provided an independent support for hypotheses on their origin. While we have unambiguously identified a range of translocations, a precise determination of breakpoint positions will have to be done using a long-read DNA sequencing technology [71,72,73].

The discrepancies in the genome structure of banana diploids observed in our study and published data then point to alternative scenarios on the origin of the important crop. The observation on structural chromosome heterozygosity confirmed the hybrid origin of cultivated banana and some of the wild diploid accessions, which were described as individual subspecies, and serves to inform breeders on possible causes of reduced fertility. Due to aberrant meiosis, gametes produced by structural heterozygotes could be aneuploid and have unbalanced chromosome translocations, duplications, and deletions. Thus, the knowledge of genome structure at the chromosomal level and the identification of structural chromosome heterozygosity will aid breeders in selecting parents for improvement programs.

## 4. Materials and Methods

### 4.1. Plant Material and Diversity Tree Construction

Representatives of twenty species and clones from the section Eumusa of genus *Musa* were obtained as in vitro rooted plants from the International *Musa* Transit Center (ITC, Bioversity International, Leuven, Belgium). In vitro plants were transferred to soil and kept in a heated greenhouse. Table 1 lists the accessions used in this work. Genetic diversity analysis of banana accessions used in the study was performed using SSR data according to Christelová et al. [28]. A Dendrogram was constructed based on the results of unweighted pair group method with arithmetic mean (UPGMA) analysis implemented in DARwin software v6.0.021 [74] and visualized in FigTree v1.4.0 [75].

### 4.2. Preparation of Oligo Painting Probes and Mitotic Metaphase Chromosome Spreads

Chromosome-arm-specific painting probes were prepared as described by Šimoníková et al. [38]. Briefly, sets of 20,000 oligomers (45 nt) covering individual chromosome arms were synthesized as immortal libraries by Arbor Biosciences (Ann Arbor, MI, USA) and then labeled directly by CY5 fluorochrome, or by digoxigenin or biotin according to Han et al. [46]. N.B.: In the reference genome assembly of *M. acuminata* “DH Pahang” [76], pseudomolecules 1, 6, 7 are oriented inversely to the traditional way karyotypes are presented, where the short arms are on the top (Appendix A, see also Šimoníková et al. [38]).

Actively growing root tips from 3 to 5 individual plants representing each genotype were collected and pre-treated in 0.05% (*w*/*v*) 8-hydroxyquinoline for three hours at room temperature, fixed in 3:1 ethanol:acetic acid fixative overnight at −20 °C and stored in 70% ethanol at −20 °C. After washing in 75 mM KCl and 7.5 mM EDTA (pH 4), root tip segments were digested in a mixture of 2% (*w*/*v*) cellulase and 2% (*w*/*v*) pectinase in 75 mM KCl and 7.5 mM EDTA (pH 4) for 90 min at 30 °C. The suspension of protoplasts thus obtained was filtered through a 150 μm nylon mesh, pelleted and washed in 70% ethanol. For further use, the protoplast suspension was stored in 70% ethanol at −20 °C. Mitotic metaphase chromosome spreads were prepared by a dropping method from protoplast suspension according to Doležel et al. [77], the slides were postfixed in 4% (*v*/*v*) formaldehyde solution in 2 × SSC (saline-sodium citrate) solution, air dried and used for FISH.

### 4.3. Fluorescence In Situ Hybridization and Image Analysis

Fluorescence in situ hybridization and image analysis were performed according to Šimoníková et al. [38]. Hybridization mixture containing 50% (*v*/*v*) formamide, 10% (*w*/*v*) dextran sulfate in 2 × SSC, and 10 ng/µL of labeled probes was added onto a slide and denatured for 3 min at 80 °C. Hybridization was carried out in a humid chamber overnight at 37 °C. The sites of hybridization of digoxigenin- and biotin-labeled probes were detected using anti-digoxigenin-FITC (Roche Applied Science, Penzberg, Germany) and streptavidin-Cy3 (ThermoFisher Scientific/Invitrogen, Carlsbad, CA, USA), respectively. Chromosomes were counterstained with DAPI and mounted in Vectashield Antifade Mounting Medium (Vector Laboratories, Burlingame, CA, USA). The slides were examined with Axio Imager Z.2 Zeiss microscope (Zeiss, Oberkochen, Germany) equipped with Cool Cube 1 camera (Metasystems, Altlussheim, Germany) and appropriate optical filters. The capture of fluorescence signals, merging of the layers, and measurement of chromosome length were performed with ISIS software 5.4.7 (Metasystems). The final image adjustment and creation of ideograms were done in Adobe Photoshop CS5. Different probe combinations hybridizing a minimum of ten preparations with mitotic metaphase chromosome spreads were used for the final karyotype reconstruction of each genotype. A minimum of ten mitotic metaphase plates per slide were captured for each probe combination.

## Figures and Tables

**Figure 1 ijms-21-07915-f001:**
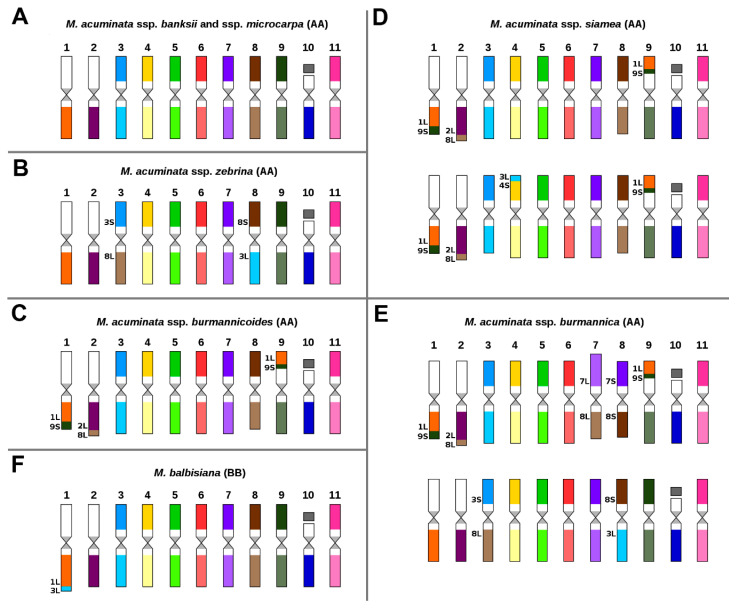
Ideograms of six diploid (2n = 2x = 22) subspecies of *Musa acuminata* and diploid (2n = 2x = 22) *Musa balbisiana*: (**A**) *M. acuminata* ssp. *banksii* “Banksii” and ssp. *microcarpa* “Borneo”; (**B**) *M. acuminata* ssp. *zebrina* “Maia Oa”; (**C**) *M. acuminata* ssp. *burmannicoides* “Calcutta 4”; (**D**) *M. acuminata* ssp. *siamea* “Pa Rayong”; (**E**) *M. acuminata* ssp. *burmannica* “Tavoy”; (**F**) *M. balbisiana* “Pisang Klutuk Wulung”. Chromosome paints were not used for the short arms of chromosomes 1, 2, and 10. Chromosomes are oriented with their short arms on top and the long arms on the bottom in all ideograms. For better orientation, translocated parts of the chromosomes contain extra labels, which include chromosome number and the chromosome arm that was involved in the rearrangement.

**Figure 2 ijms-21-07915-f002:**
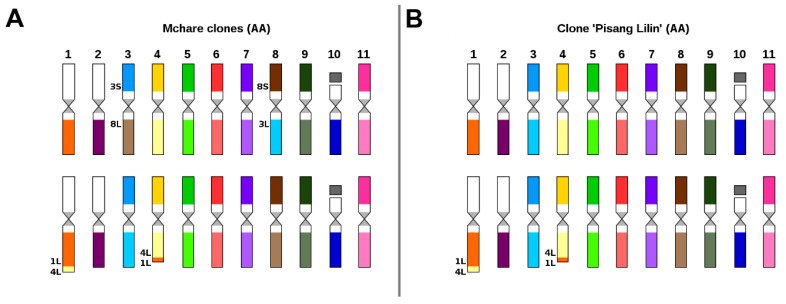
Ideograms of diploid (2n = 2x = 22) *Musa* clones: (**A**) *M. acuminata* subgr. Mchare, clones “Huti white”, “Huti (Shumba nyeelu)”, and “Ndyali”; (**B**) *M. acuminata* clone “Pisang Lilin”. Chromosome paints were not used for the short arms of chromosomes 1, 2, and 10. Chromosomes are oriented with their short arms on top and the long arms on the bottom in all ideograms. For better orientation, translocated parts of the chromosomes contain extra labels, which include chromosome number and the chromosome arm that was involved in the rearrangement.

**Figure 3 ijms-21-07915-f003:**
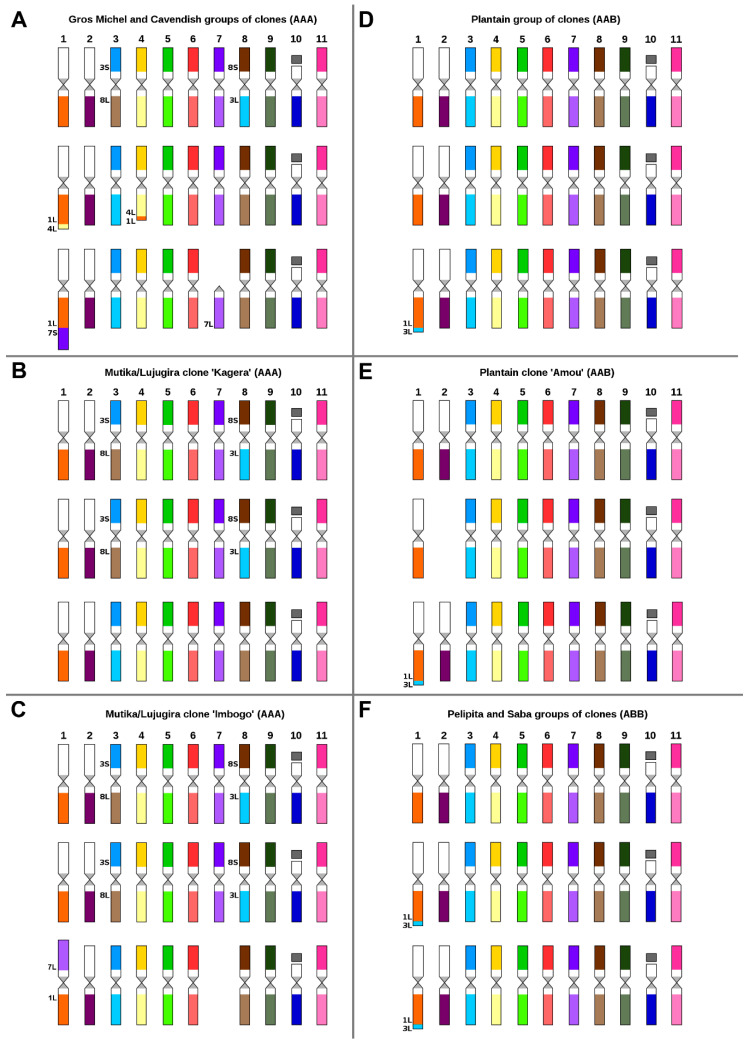
Ideograms of triploid (2n = 3x = 33) *Musa* accessions: (**A**) Clones “Gros Michel” and “Poyo”; (**B**) East African Highland banana (EAHB) clone “Kagera”; (**C**) aneuploid EAHB clone “Imbogo” (2n = 3x − 1 = 32); (**D**) plantains “3 Hands Planty” and “Obino l’Ewai”; (**E**) aneuploid plantain clone “Amou” (2n = 3x − 1 = 32); (**F**) clones “Pelipita” and “Saba sa Hapon”. Chromosome paints were not used for the short arms of chromosomes 1, 2, and 10. Chromosomes are oriented with their short arms on top and the long arms on the bottom in all ideograms. For better orientation, translocated parts of the chromosomes contain extra labels, which include chromosome number and the chromosome arm that was involved in the rearrangement.

**Figure 4 ijms-21-07915-f004:**
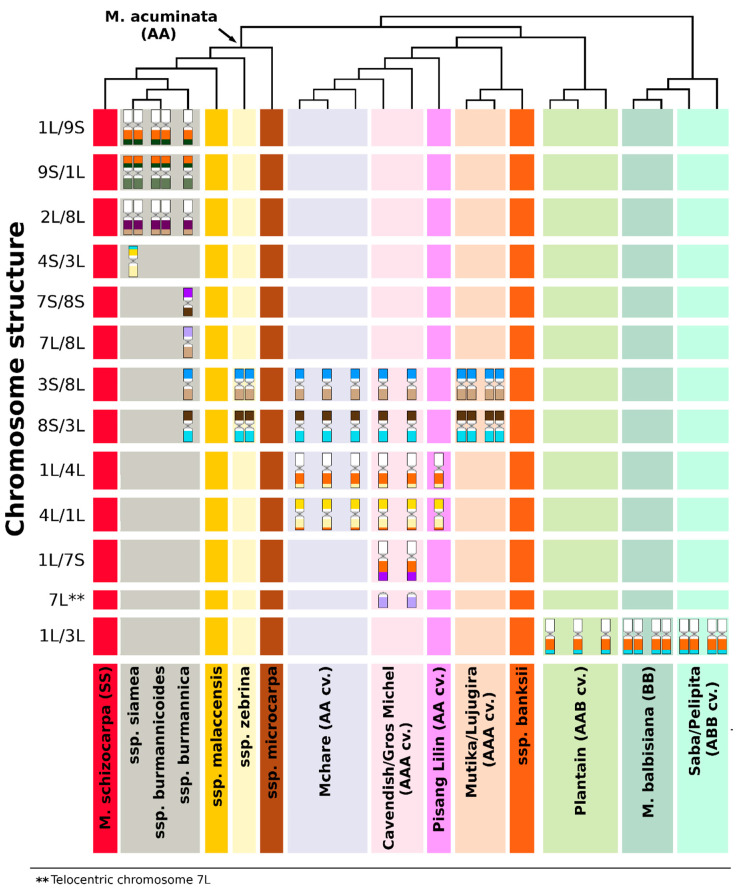
Overview of common translocation events revealed by oligo painting fluorescence in situ hybridization (FISH) in *Musa*: diversity tree, constructed using simple sequence repeat (SSR) markers according to Christelová et al. [28], shows the relationships among *Musa* species, subspecies, and hybrid clones. Lineages of closely related accessions and groups of edible banana clones are highlighted in different colors. Individual chromosome structures (displayed as chromosome schemes) are depicted in rows, and their number correspond to the number of chromosomes bearing the rearrangement in the nuclear genome in somatic cell lines (2n).

**Table 1 ijms-21-07915-t001:** List of Musa accessions analyzed in this work and their genomic constitution.

Species	Subspecies/Subgroup	Accession Name	ITC Code ^a^	Genomic Constitution	Chromosome Number (2n)
*M. acuminata*	*banksii*	Banksii	0806	AA	22
	*microcarpa*	Borneo	0253	AA	22
	*zebrina*	Maia Oa	0728	AA	22
	*burmannica*	Tavoy	0072	AA	22
	*burmannicoides*	Calcutta 4	0249	AA	22
	*siamea*	Pa Rayong	0672	AA	22
*M. balbisiana*	*-*	Pisang Klutuk Wulung	*-*	BB	22
Cultivars	unknown	Pisang Lilin	*-*	AA	22
	Mchare	Huti White	*-*	AA	22
	Mchare	Huti (Shumba nyeelu)	1452	AA	22
	Mchare	Ndyali	1552	AA	22
	Cavendish	Poyo	1482	AAA	33
	Gros Michel	Gros Michel	0484	AAA	33
	Mutika/Lujugira	Imbogo	0168	AAA	32*
	Mutika/Lujugira	Kagera	0141	AAA	33
	Plantain (Horn)	3 Hands Planty	1132	AAB	33
	Plantain (French)	Amou	0963	AAB	32 *
	Plantain (French)	Obino l’Ewai	0109	AAB	33
	Pelipita	Pelipita	0472	ABB	33
	Saba	Saba sa Hapon	1777	ABB	33

^a^ Code assigned by the International Transit Center (ITC, Leuven, Belgium). * Aneuploidy observed in our study.

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
