# Peer review of "Chromosome Painting in Cultivated Bananas and Their Wild Relatives (Musa spp.) Reveals Differences in Chromosome Structure"

_ijms, 2020, doi:10.3390/ijms21217915_

Round 1

Reviewer 1 Report

The presented article contains a description of the evolution of the karyotype of a cultivated banana. The presented material sheds new light on the evolution of this important plant. However, the Results section is not entirely clearly written, and I recommend revising it in accordance with the following considerations.

  1. It is necessary to describe the results in a more structured manner. The figures are now described out of order (fig. 1A, 1C, 3A...). It is inconvenient to jump from one panel to another. And I'm not sure if all the microscopic images are neccessary for manuscript logic. They can be moved to the Supplementary, and idiograms can be added to the text (they better demonstrate translocations).
  2. Some of the data is provided only in the Supplementary. I had to regularly look into the application, although there should be either repetitions or unimportant data.

And while reading the article, I had the following question. The Introduction mentions that the banana subspecies have different geographic distributions. Doesn't it make sense to present the results of the study in the form of a map on which to show the localization of variants with different types of chromosomal rearrangements? The Introduction discusses geographic issues in detail, so this question arises at the end of the article.

I have no more concern about this manuscript, I can recommend this article for publication in International Journal of Molecular Sciences after minor revision.

Author Response

We appreciate the positive comments and suggestions on how to improve our manuscript. We have followed all suggestions and we have modified our manuscript accordingly.

1) It is necessary to describe the results in a more structured manner. The figures are now described out of order (fig. 1A, 1C, 3A...). It is inconvenient to jump from one panel to another. And I'm not sure if all the microscopic images are necessary for manuscript logic. They can be moved to the Supplementary, and idiograms can be added to the text (they better demonstrate translocations).

Response: We appreciate the suggestion and we have modified our manuscript as suggested. We prepared new figures (Figure 1, Figure 2, Figure 3) showing the idiograms of the analyzed accessions, and included them in main the body of the manuscript. The original figures, which include examples of microscopic images, were moved to the Supplements.

2) Some of the data is provided only in the Supplementary. I had to regularly look into the application, although there should be either repetitions, or unimportant data.

Response: We agree, that e.g. Supplementary Table S1 provides the same information as shown in the idiograms. However, we believe that readers will appreciate a Table summarizing all results on one page.

3) The Introduction mentions that the banana subspecies have different geographic distributions. Doesn't it make sense to present the results of the study in the form of a map on which to show the localization of variants with different types of chromosomal rearrangements? The Introduction discusses geographic issues in detail, so this question arises at the end of the article.
Response: We appreciate this suggestion. However, we do not think that inclusion of a map would provide information relevant to the subject of the paper. The information on geographical distribution has been included in the Introduction because, in addition to plant morphology and basic chromosome number, it was used to characterize species and subspecies of Musa. To provide an overview of different chromosome translocations (identified in comparison to the reference genome sequence of M. acuminata ssp. malaccensis), we created Figure 4, which shows common translocation events in Musa related to the evolutionary relationships among Musa species, subspecies and hybrid clones.

Reviewer 2 Report

It was my honour to read this technically brilliant piece of work. I did not find any weak point of the chains consisting of presumption-experiment-result-conclusion in this manuscript. All results are clearly demonstrated and the data are well and still critically summarized.

My only suggestion is about that general statement about "knowledge on genome structure at chromosomal level... will aid breeders in selecting parents for improvement programs".

Would it be possible that readers get a clearer picture of how it may aid? Is there any specific example of parents with an advantageous trait that will be usable in an implied strategy of breeding?

Nice MS. Thank you.

Author Response

We appreciate the positive comments and suggestions on how to improve our manuscript. We have followed all suggestions and we have modified our manuscript accordingly.

1) My only suggestion is about that general statement about "knowledge on genome structure at chromosomal level... will aid breeders in selecting parents for improvement programs". Would it be possible that readers get a clearer picture of how it may aid? Is there any specific example of parents with an advantageous trait that will be usable in an implied strategy of breeding?

Response: We apologize for not explaining clearly a need for understanding genome organization at chromosomal level. A hybrid originating from a cross between two accession differing in chromosome structure, i.e. in the presence of different translocations, will be structurally heterozygous. Due to aberrant meiosis, gametes produced by a structural heterozygote could be aneuploid and have unbalanced chromosome translocations, duplications and deletions. We have modified the text to explain this fact.

Reviewer 3 Report

To analyze the domestication of banana chromosomes and karyotypes, the authors build on their previous publication (Šimoníková,  D.;  Němečková,  A.;  Karafiátová,  M.;  Uwimana,  B.;  Swennen,  R.;  Doležel,  J.;  Hřibová,  E. Chromosome  painting  facilitates  anchoring  reference  genome  sequence  to  chromosomes  in  situ  and integrated karyotyping in banana (Musa spp.). Front. Plant Sci. 2019, 10, 1503). Here, they apply the developed oligonucleotide (oligo)-FISH probes for chromosome painting of twenty Musa cultivars, laying bare countless previously unknown chromosome mutations, especially translocations, with implications for banana breeding and selection.

This is an exemplary and comprehensive cytogenetic work with high quality figures, and I recommend additional hightlighting/promotion by the journal.

The cytogenetics figures are of high quality and also the summarizing figures/tables, Fig. 5 and Table S1, are well prepared and extremely helpful to the reader. The introduction is well-written and provides all necessary facts. The methods are comprehensive. The discussion is in line with the results.  I also thought that it was interesting that the authors compared their findings to recent sequencing efforts, e.g. line 320.

I only have one minor remark regarding readability of the figures: Currently, the text, the figures, the figure legend, and the accession list (Table 1) is needed to understand each(!) panel. This is extremely time-consuming for the reader. The manuscript would greatly benefit from:

  • An inclusion of the subspecies, accession name, and genomic constitution directly into the panels
  • The inclusion of the subspecies also into the figure legend

I recommend acceptance after minor revision.

Author Response

We appreciate the positive comments and suggestions on how to improve our manuscript. We have followed all suggestions and we have modified our manuscript accordingly.

1) I only have one minor remark regarding readability of the figures: Currently, the text, the figures, the figure legend, and the accession list (Table 1) is needed to understand each(!) panel. This is extremely time-consuming for the reader. The manuscript would greatly benefit from:

An inclusion of the subspecies, accession name, and genomic constitution directly into the panels The inclusion of the subspecies also into the figure legend  

Response: We appreciate the comments and recommendations and we have modified the figures accordingly. Following the suggestion of Reviewer 1, we moved microscopic images to the Supplements. Based on your recommendation, we added the accession name, subspecies or group of edible clones, and genome constitution in the individual figures of the panels.